# Associations between Macronutrient Intakes and Obesity/Metabolic Risk Phenotypes: Findings of the Korean National Health and Nutrition Examination Survey

**DOI:** 10.3390/nu11030628

**Published:** 2019-03-14

**Authors:** Ha-Na Kim, Sang-Wook Song

**Affiliations:** Department of Family Medicine, St. Vincent’s Hospital, College of Medicine, The Catholic University of Korea, Seoul 06591, Korea; onef01@catholic.ac.kr

**Keywords:** dietary carbohydrates, dietary proteins, dietary fats, metabolic syndrome, obesity

## Abstract

Obesity is a risk factor for many health issues, as are metabolic abnormalities. However, few studies have addressed the associations between obesity/metabolic risk phenotypes and dietary macronutrient intakes (carbohydrate, protein, and fat). Therefore, this study examined the associations between macronutrient intakes and obesity/metabolic risk phenotypes in a Korean population. We used data from the Korean National Health and Nutrition Examination Survey, a cross-sectional survey of Korean civilians, conducted in 2014 and 2016, and data on a total of 7374 participants were analyzed. Macronutrient intakes were defined as the proportions of energy derived from carbohydrate, protein, and fat. Those exhibiting obesity/metabolic risk phenotypes (or not) were divided into four groups: normal weight without metabolic abnormalities; obese without metabolic abnormalities; normal weight with metabolic abnormalities; and obese with metabolic abnormalities. After adjusting for age, smoking status, alcohol consumption, extent of physical activity, household income, and daily fiber intake, no association was found between the proportions of carbohydrate, protein, or fat intakes and obesity/metabolic risk phenotypes except for a positive association between metabolically healthy but obese status and low protein intake in females. Further studies are required to evaluate the effects of macronutrient intakes on obesity/metabolic risk phenotypes and associated health outcomes.

## 1. Introduction

Obesity, the accumulation of excess body fat, is an important public health problem associated with an increased risk of type 2 diabetes mellitus, dyslipidemia, hypertension, and cardiovascular disease (CVD); the prevalence of obesity is increasing worldwide [1]. However, not all obese individuals exhibit metabolic abnormalities such as elevated fasting glucose levels, insulin resistance, increased triglyceride (TG) levels, high blood pressure, or low levels of high-density lipoprotein cholesterol (HDL-C), all of which compromise health. Furthermore, not all normal-weight individuals exhibit favorable metabolic profiles. Therefore, combinations of obesity with metabolic abnormalities (the obesity/metabolic risk phenotypes) have attracted increasing attention [2,3]. These phenotypes can be divided into four: metabolically healthy and of normal weight (MHNW); metabolically healthy but obese (MHO; a favorable metabolic profile despite excessive body fat); metabolically abnormal but of normal weight (MANW; exhibiting metabolic risk factors despite a normal body mass index (BMI)); and metabolically abnormal and obese (MAO; obese and exhibiting metabolic risk factors).

Age, sex, the extent of physical activity, alcohol intake, and concomitant disease influence the obesity/metabolic risk phenotypes [4,5,6,7,8]. However, few clinical studies have addressed the associations between obesity/metabolic risk phenotypes and dietary macronutrient intakes (carbohydrate, protein, and fat) [9], although several studies have found associations between macronutrient intakes and both obesity [10,11,12] and metabolic abnormalities [13,14,15,16]. Therefore, using data from the Korean National Health and Nutrition Examination Survey (KNHANES), we investigated whether dietary carbohydrate, protein, and fat intakes were associated with obesity/metabolic risk phenotypes in Korean adults.

## 2. Materials and Methods

### 2.1. Study Population

We used KNHANES data collected in 2014 and 2016. The KNHANES is performed by the Korean Centers for Disease Control and Prevention at 3-year intervals to assess public health status and to provide baseline data for the development, establishment, and evaluation of Korean public health policies. All KNHANES subjects are non-institutionalized and aged ≥1 year; they are selected using a stratified, multi-stage, cluster probability sampling design to ensure that the sample is independent, homogeneous, and nationally representative. Data are collected using household interviews, via anthropometric and biochemical measurements, and employing nutritional assessment. All protocols in play have been approved by the Institutional Review Board of the Korean Centers for Disease Control and Prevention, and all participants provided written informed consent at baseline.

In this cross-sectional study, we originally evaluated data from 17,223 adults, ≥20 years of age, selected from 21,948 KNHANES participants. We excluded participants for whom important data were missing, thus data required for classification of obesity/metabolic risk phenotypes (*n* = 778); those who reported implausible energy intakes (<500 or >5000 kcal/day; *n* = 2477); those who were on specific diets (*n* = 3442); and those for whom food intake on the day of the 24-h dietary recall survey was not representative of their usual intake (*n* = 3152). Thus, data from 7374 participants were analyzed (Figure 1). The study was approved by the Institutional Review Board of the Catholic University of Korea (IRB approval number: VC18ZESI0196).

### 2.2. Classification of Obesity/Metabolic Risk Phenotypes

BMI was classified as normal or obese (BMI < 25 kg/m^2^ and ≥ 25.0 kg/m^2^, respectively) [17]. Metabolically abnormal status was defined when three or more of the following features were present, consistent with the criteria used by the revised National Cholesterol Education Program Adult Treatment Panel III to define metabolic syndrome (MetS) [18]: waist circumference (WC) ≥ 90 cm for males (≥85 cm for females) [19]; TG level ≥ 150 mg/dL or the use of medication to decrease the TG level; HDL-C level <40 mg/dL for males (<50 mg/dL for females) or prescription of medication to lower the HDL-C level; systolic blood pressure ≥130 mmHg, diastolic blood pressure ≥85 mmHg, or the use of an anti-hypertensive agent; and fasting glucose level ≥100 mg/dL or prescription of a blood glucose-lowering agent. The subjects were divided into four groups based on the presence/absence of obesity and MetS status: MHNW, MHO, MANW, and MAO.

### 2.3. Dietary Assessments

Trained interviewers estimated dietary intakes via 24-h dietary recall of all foods and beverages. The survey was performed at participants’ homes, and tools such as food models, two-dimensional food photographs, and containers were used to help participants recall nutrient intakes. Dietary intakes were estimated using the food composition tables of the Korean Rural Development Administration, in combination with the nutrient database of the Korean Health and Industry of Development Institute [20]. Calculations of energy derived from carbohydrate, protein, and fat employed the standard conversion factors for grams to kilocalories (4 kcal/g for carbohydrate and protein, 9 kcal/g for fat) and the proportions of energy derived from macronutrients were then calculated as energies imparted by a particular macronutrient/total energy intake (%). High or low intakes of carbohydrate, protein, and fat were those that were more or less, respectively, than the recommended intakes for Korean adults; these are 55–65, 7–20, and 15–30% of calories from carbohydrate, protein, and fat, respectively [21].

### 2.4. Laboratory and Anthropometric Measurements

Blood samples were collected from the antecubital vein after at least 12 h of fasting, processed, immediately refrigerated, and transported in the cold to the Central Testing Institute in Seoul, Korea. All blood samples were analyzed within 24 h of arrival. Fasting plasma glucose, TG, and HDL-C levels were measured with the aid of an auto-analyzer (Hitachi Automatic Analyzer 7600, Hitachi, Japan). Anthropometric measurements were performed by trained examiners. Height and weight were measured after an overnight fast with participants wearing lightweight gowns, and WC was measured (using a measuring tape) in the horizontal plane around the umbilical region after exhalation. Blood pressure measurements were taken in the sitting position after a rest period of at least 5 min. BMI was calculated as weight (in kilogram) divided by the square of height (in meter).

### 2.5. Other Variables

Self-reported age, sex, smoking status, alcohol consumption, extent of physical activity, and household income data were recorded. Cigarette smoking was divided into three categories based on current use: non-smoker, ex-smoker, and current smoker. Information on alcohol consumption included the frequency of drinking days and number of drinks consumed per day during the 1 year that preceded the KNHANES household interview. We used the Korean version of a “standard drink” (any drink that contains 10 g pure alcohol) based on alcohol contents of 4.5 vol% for beer, 12 vol% for wine, 6 vol% for Korean traditional makgeolli, 21 vol% for Korean soju, and 40 vol% for whisky; alcohol consumption was classified into three categories: abstinence (no alcoholic drinks consumed within the last year), moderate drinking (≤14 standard drinks consumed by males and ≤7 by females per week), and heavy drinking (>14 standard drinks consumed by males and >7 by females per week) [22]. Participants were asked about their extent of physical activity during the week that preceded the interview; this was classified as low or not. Low-level activity was defined as ≤150 min of moderate-intensity or ≤75 min of vigorous exercise per week [23]. Household income was divided into monthly equivalent household values (in quartiles), estimated as total income divided by the square root of the number of household members.

### 2.6. Statistical Analysis

We used the SAS PROC SURVEY module (which considers strata, clusters, and weights) to analyze the data; we employed a complex sampling design. All analyses were performed using the sample weightings of KNHANES. Sex-specific features were evaluated via analysis of variance for continuous variables and with the aid of the chi-squared test for dichotomous variables. Data are expressed as means ± standard errors or as percentages. 

Differences in mean macronutrient intakes by obesity/metabolic risk phenotype were evaluated using analysis of covariance; all of age; smoking status; alcohol consumption; physical activity level; household income; and daily fiber, carbohydrate, protein, and fat intakes (except for models evaluating macronutrients individually) served as covariates. 

The associations between obesity/metabolic risk phenotypes and high or low macronutrient intakes were analyzed via multiple logistic regression after adjusting for covariates. All statistical analyses were performed using SAS software (ver. 9.2; SAS Institute, Cary, NC, USA). A *p* value < 0.05 was considered to reflect significance.

## 3. Results

### 3.1. Participant Characteristics by Obesity/Metabolic Risk Phenotypes

Data from 7374 participants (3296 males and 4078 females) were collected. The prevalences of male MHNW, MHO, MANW, and MAO status were 53.1, 16.3, 11.0, and 19.6%, respectively; the figures for females were 55.8, 9.5, 17.4, and 17.3%, respectively. The obesity subtypes varied significantly by age; physical activity level; household income; any history of diabetes, hypertension, or dyslipidemia; BMI; WC; systolic and diastolic blood pressures; serum fasting glucose level; and lipid profile (in both males and females; Table 1). 

### 3.2. Mean Macronutrient Intakes by Obesity/Metabolic Risk Phenotypes

The mean macronutrient intake proportions by obesity/metabolic risk phenotypes are shown in Table 2. MANW males obtained a higher proportion of energy from carbohydrate and lower proportions from fat and protein than did the other groups. In females, the proportion of carbohydrate intake increased and the proportions of fat and protein intake decreased according to obesity/metabolic risk phenotypes. However, macronutrient intake proportions did not differ significantly, and exhibited no trend by the obesity/metabolic risk phenotypes, in either males or females, after adjustment for age; smoking status; alcohol consumption; physical activity level; household income; or daily fiber, carbohydrate, protein, or fat intake (except in models used to evaluate the effects of individual macronutrients).

### 3.3. Associations between High or Low Macronutrient Intakes and the Obesity/Metabolic Risk Phenotypes

Unadjusted odds ratios (ORs), age-adjusted ORs (Model 1), and multivariate-adjusted ORs (Models 2 and 3) for high or low macronutrient intakes by obesity/metabolic risk phenotypes are shown in Table 3 and Table 4. MHO males were more likely to have a low carbohydrate intake compared to MHNW controls (unadjusted OR 1.41, 95% confidence interval (CI) 1.04–1.92). The unadjusted ORs for high and low carbohydrate intakes by MANW males were 1.39 (95% CI 1.05–1.82) and 0.58 (95% CI 0.39–0.86), respectively. The unadjusted ORs for high and low fat intakes by MANW males were 0.58 (95% CI 0.37–0.91) and 1.39 (95% CI 1.09–1.78), respectively. For MAO males, the unadjusted OR for high fat intake was 0.66 (95% CI 0.47–0.93). However, these associations did not remain significant after adjustment for covariates (Table 3).

Of females, both the MANW and MAO groups were more likely to have high carbohydrate and low fat intakes compared to MHNW controls. High-carbohydrate/low-fat intake females exhibited an increasing trend in terms of the obesity/metabolic risk phenotypes (both p values for trend < 0.001), and a decreasing trend for high-fat/low-carbohydrate females (both p values for trend < 0.001), but these associations disappeared after adjusting for all covariates. No association was evident between protein intake and obesity/metabolic risk phenotypes except for a positive association between female MHO status and low protein intake (multivariate-adjusted OR 5.85, 95% CI 1.13–30.31) (Table 4).

## 4. Discussion

We investigated the associations between dietary carbohydrate, protein, and fat intakes and obesity/metabolic risk phenotypes; we thus considered body fat and metabolic risks simultaneously. We found no association between the intake of any macronutrient and any obesity/metabolic risk phenotype except for a positive relationship between low protein intake and MHO status in females, after adjusting for the abovementioned covariates.

Obesity triggers inflammation and oxidative stress, increasing the risk of metabolic abnormalities. However, metabolically healthy but obese, and metabolically unhealthy but non-obese, individuals are often encountered; obesity/metabolic risk phenotypes remain challenging in clinical practice. Although the results were controversial [24,25], obesity/metabolic risk phenotypes predicted the risks of CVD and all-cause mortality (especially of MAO subjects) relatively well [26,27]. Thus, definition of obesity subtypes by reference to metabolic risk status, and clarification of the factors influencing such subtypes, may facilitate personalized management (early medical treatment combined with lifestyle modifications).

Both high and low macronutrient intakes were associated with body fat levels and obesity in previous studies [28,29,30], suggesting that variations in the levels of carbohydrate, protein, and fat consumed may affect obesity. In addition, various dietary macronutrients were associated with metabolic abnormalities that in turn increased the risk of CVD and mortality. High carbohydrate intake was associated with a reduced HDL-C and an elevated TG level in Korean adults [13]. A high-protein diet reduced total cholesterol and TG levels [15,31]; as fat intake increased, the levels of low-density lipoprotein cholesterol (LDL-C) and TG tended to rise [16]. Therefore, identification of how macronutrient intakes may be used to stratify obesity in terms of metabolic abnormalities (thus, to define obesity/metabolic risk phenotypes) is important to ensure specific (individual) approaches to patients of various obesity subtypes. However, only one prior study explored whether the proportions of energy from carbohydrate, protein, or fat influenced the obesity/metabolic risk phenotype. In Korean adults, high-carbohydrate and high-protein diets were associated with an increased and decreased risk of MANW status, respectively, compared to low-carbohydrate and low-protein diets [9], in contrast to the results of this study.

The absence of an association between macronutrient intakes and obesity/metabolic risk phenotypes is amenable to several explanations. In Koreans, high-carbohydrate and low-fat diets were positively associated with metabolic abnormalities and an increased prevalence of MetS [13,32], but no prior study has evaluated the associations between macronutrient intakes and body fat level/obesity and we found no association between carbohydrate intake and body fat level or obesity (unpublished data), implying that a combination of obesity and a metabolic abnormality (thus, expression of an obesity/metabolic risk phenotype) in Korean adults may attenuate the associations between macronutrient intakes and metabolic abnormalities. In addition, a meta-analysis showed that low carbohydrate diets (carbohydrate intake <45% of total energy) and isoenergetic-balanced diets (45–65% carbohydrate intake as a proportion of total energy) were not different in terms of weight loss after a 2-year follow-up, indicating that weight loss could be the result of reduced total energy intake rather than to any manipulation of macronutrient proportions [33]. However, in a randomized controlled trial, TG and C-reactive protein levels did not normalize as rapidly in metabolically abnormal subjects as in metabolically healthy subjects; the postprandial TG and CRP response of the former group was less flexible as evidenced by less improvement in the postprandial TG response and inflammation status after dynamic fat load testing, whether the subjects were obese or not [34]. In one case-control study, the levels of atherogenic LDL-C materials (such as those of small LDL particles and oxidized LDL), and inflammatory cytokines including tumor necrosis factor-α and interleukin-6 were higher in MANW females than controls [35], suggesting that metabolic abnormalities evident even in non-obese subjects can negatively affect the outcomes of type 2 diabetes mellitus, dyslipidemia, hypertension, and CVD. Further studies are needed to explore whether the intakes of macronutrients that play roles in metabolic abnormalities predict health outcomes [13,14,15,16,32]. In addition, several studies have shown that dietary carbohydrate quality represented by glycemic index or carbohydrate quality index and the amounts of polyunsaturated- and saturated fatty acids affect chronic inflammation, oxidative stress, and insulin resistance [35,36,37]. Therefore, associations between macronutrient qualities and types (thus not only quantities) and obesity/metabolic risk phenotypes require evaluation.

Our finding that MHO status in females was positively associated with low protein intake (≤7% of total energy) is notable. A higher proportion of protein in the diet may promote satiety, reduce overall energy intake, and positively influence body composition [31,38], but no previous study found an association between low protein intake and fewer metabolic abnormalities in obese subjects. Furthermore, the *p* value was 0.035, even though the number of females reporting low protein intake was relatively small (data not shown). Additional studies with larger sample sizes are warranted to clarify the association between protein intake and MHO status.

The strengths of this study are that the data are nationally representative of the South Korean population and that this is the first study to explore the associations between carbohydrate, protein, and fat intakes, on the one hand, and Korean obesity/metabolic risk phenotypes, on the other. However, the study had certain limitations. First, it was cross-sectional in design. Second, dietary intake was estimated using a 24-h recall method; diet can fluctuate on a day-to-day basis. Therefore, we excluded participants whose diets on the day of the survey were not representative of their usual intakes. Third, the qualities (i.e., types) of dietary carbohydrates, proteins, and fats consumed (of low or high glycemic load; and varying in terms of the proportions of sugar, unsaturated/saturated fatty acids, and amino acids) were not captured by the KNHANES. Fourth, we did not consider other potentially relevant factors, such as dietary or supplemental intake of vitamins and minerals that might affect obesity/metabolic risk phenotypes, due to insufficient data.

## 5. Conclusions

We found no significant association between carbohydrate, protein, or fat intakes (as proportions of energy consumed) and obesity/metabolic risk phenotypes in Korean adults, except for an association between low protein intake in females and MHO status. Further studies are needed to evaluate the effects of macronutrient intakes on obesity/metabolic risk phenotypes and associated health outcomes, with consideration of not only the quantities but also the qualities of macronutrients consumed.

## Figures and Tables

**Figure 1 nutrients-11-00628-f001:**
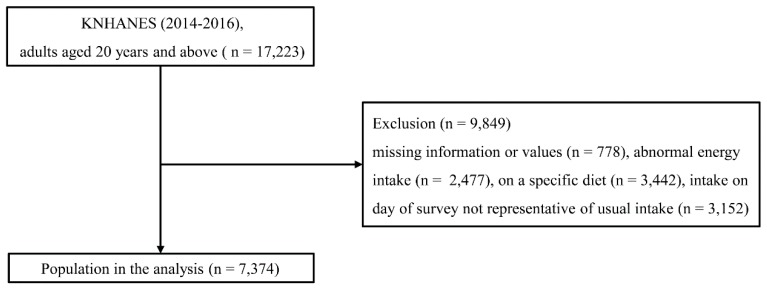
Study population: Data from the 2014–2016 Korean National Health and Nutrition Examination Survey (KNHANES).

**Table 1 nutrients-11-00628-t001:** Characteristics of study participants by obesity/metabolic risk phenotypes.

	Total	Males					Females				
		MHNW	MHO	MANW	MAO	*p* Value	MHNW	MHO	MANW	MAO	*p* Value
*N*	7374	1705	478	434	679	-	2079	386	818	795	-
Age (years)	48.7 ± 0.3	46.6 ± 0.5	43.2 ± 0.7	57.2 ± 0.9	50.4 ± 0.7	<0.001	43.8 ± 0.4	48.7 ± 0.9	59.3 ± 0.7	58.3 ± 0.6	<0.001
Current smoking (%)	21.8	38.4	36.2	41.7	37.0	0.509	4.5	5.0	4.7	5.1	0.964
Heavy drinking (%)	12.1	15.5	17.5	26.6	26.3	<0.001	4.5	6.8	4.4	4.4	0.453
Physical activity (low, %)	44.6	50.1	55.6	43.4	38.8	<0.001	45.3	42.0	33.8	34.0	<0.001
Household income (low, %)	41.9	39.3	37.4	54.3	43.2	<0.001	33.8	41.7	56.4	56.5	<0.001
Energy intake (kcal)	2038.7 ± 12.2	2362.7 ± 25.5	2331.9 ± 46.5	2201.8 ± 45.4	2298.9 ± 39.0	0.013	1778.8 ± 15.8	1761.4 ± 38.8	1661.1 ± 25.9	1670.6 ± 26.0	<0.001
Carbohydrate intake (g)	316.1 ± 1.8	355.1 ± 3.8	339.8 ± 6.3	334.3 ± 7.0	339.9 ± 5.2	0.007	282.2 ± 2.7	282.4 ± 6.4	286.1 ± 4.5	285.6 ± 4.6	0.830
Fat intake (g)	43.6 ± 0.5	51.0 ± 0.9	54.9 ± 2.0	41.1 ± 1.6	47.8 ± 1.5	<0.001	41.1 ± 0.7	37.9 ± 1.5	29.7 ± 1.1	30.4 ± 0.9	<0.001
Protein intake (g)	70.2 ± 0.6	81.3 ± 1.2	84.5 ± 2.3	73.5 ± 1.8	80.6 ± 1.8	<0.001	61.3 ± 0.7	61.7 ± 2.1	55.1 ± 1.1	55.2 ± 1.2	<0.001
Fiber intake (g)	24.0 ± 0.2	25.6 ± 0.4	24.6 ± 0.7	26.0 ± 0.7	25.9 ± 0.6	0.431	22.1 ± 0.3	22.7 ± 0.7	23.2 ± 0.5	22.3 ± 0.5	0.297
Medical history (%)											
Diabetes	8.9	4.6	4.2	27.3	20.2	<0.001	1.6	3.7	14.5	22.6	<0.001
Hypertension	26.9	16.0	20.1	58.0	53.9	<0.001	10.1	18.8	43.2	57.5	<0.001
Dyslipidemia	16.1	9.4	12.2	24.7	23.3	<0.001	10.8	19.8	25.2	32.5	<0.001
Waist circumference (cm)	81.9 ± 0.2	79.9 ± 0.2	90.3 ± 0.3	85.1 ± 0.3	95.9 ± 0.3	<0.001	72.9 ± 0.2	86.8 ± 0.4	79.4 ± 0.3	90.9 ± 0.3	<0.001
Body mass index (kg/m^2^)	23.7 ± 0.1	22.1 ± 0.1	26.9 ± 0.1	23.0 ± 0.1	28.1 ± 0.1	<0.001	21.1 ± 0.1	27.1 ± 0.1	22.6 ± 0.1	28.0 ± 0.1	<0.001
SBP (mmHg)	118.1 ± 0.3	116.2 ± 0.4	119.8 ± 0.6	127.8 ± 0.8	127.4 ± 0.6	<0.001	109.6 ± 0.3	116.5 ± 0.9	123.7 ± 0.8	127.1 ± 0.7	<0.001
DBP (mmHg)	75.5 ± 0.2	75.5 ± 0.3	78.2 ± 0.5	79.7 ± 0.6	83.2 ± 0.4	<0.001	70.8 ± 0.2	74.8 ± 0.5	74.7 ± 0.4	77.7 ± 0.4	<0.001
Fasting glucose (mL/dL)	99.4 ± 0.3	96.1 ± 0.5	97.5 ± 0.8	117.7 ± 1.9	110.9 ± 1.1	<0.001	91.2 ± 0.3	95.8 ± 1.1	103.9 ± 0.9	109.9 ± 1.2	<0.001
Total cholesterol (mg/dL)	190.4 ± 0.5	185.8 ± 0.9	195.1 ± 1.7	187.4 ± 2.1	196.5 ± 1.9	<0.001	188.9 ± 0.9	201.5 ± 2.0	187.9 ± 1.4	195.9 ± 1.6	<0.001
LDL-C (mg/dL)	115.5 ± 0.7	113.7 ± 1.3	124.5 ± 2.1	105.6 ± 2.6	119.3 ± 1.9	<0.001	110.1 ± 1.2	123.1 ± 3.1	115.9 ± 1.8	122.8 ± 2.0	<0.001
HDL-C (mg/dL)	50.8 ± 0.2	51.1 ± 0.3	46.9 ± 0.4	41.2 ± 0.5	41.3 ± 0.4	<0.001	59.9 ± 0.3	58.1 ± 0.5	43.6 ± 0.3	44.9 ± 0.4	<0.001
Triglycerides (mg/dL)	141.2 ± 1.9	125.2 ± 3.7	142.9 ± 5.4	251.6 ± 11.3	248.7 ± 8.2	<0.001	83.6 ± 1.0	108.1 ± 3.3	162.7 ± 4.2	172.7 ± 3.9	<0.001

Values are means ± standard errors or percentages. Abbreviations: SBP, systolic blood pressure; DBP, diastolic blood pressure; HDL-C, high-density lipoprotein cholesterol; LDL-C, low-density lipoprotein cholesterol; MHNW, metabolically healthy and of normal weight; MHO, metabolically healthy but obese; MANW, metabolically abnormal but of normal weight; MAO, metabolically abnormal and obese.

**Table 2 nutrients-11-00628-t002:** Mean proportions of macronutrient intakes by obesity/metabolic risk phenotypes.

		MHNW	MHO	MANW	MAO	*p* Value	*p* for Trend
Males							
Unadjusted	Carbohydrate (%)	65.5 ± 0.3	63.1 ± 0.6	67.8 ± 0.6	65.6 ± 0.5	<0.001	0.250
	Fat (%)	19.9 ± 0.3	21.7 ± 0.5	17.6 ± 0.5	19.3 ± 0.4	<0.001	0.017
	Protein (%)	14.7 ± 0.1	15.2 ± 0.2	14.6 ± 0.2	15.2 ± 0.2	0.025	0.069
Model 1	Carbohydrate (%)	65.8 ± 0.3	64.4 ± 0.5	65.1 ± 0.5	64.8 ± 0.4	0.074	0.044
	Fat (%)	19.5 ± 0.2	20.5 ± 0.4	19.8 ± 0.4	19.8 ± 0.3	0.345	0.416
	Protein (%)	14.5 ± 0.1	15.0 ± 0.2	15.0 ± 0.2	15.2 ± 0.2	0.022	0.003
Model 2	Carbohydrate (%)	64.9 ± 0.3	63.5 ± 0.6	64.5 ± 0.5	64.2 ± 0.4	0.124	0.126
	Fat (%)	19.8 ± 0.2	20.7 ± 0.5	20.0 ± 0.4	20.0 ± 0.4	0.347	0.492
	Protein (%)	15.2 ± 0.1	15.6 ± 0.2	15.4 ± 0.2	15.7 ± 0.2	0.117	0.030
Model 3	Carbohydrate (%)	64.3 ± 0.2	64.2 ± 0.3	63.9 ± 0.3	64.1 ± 0.3	0.542	0.280
	Fat (%)	18.5 ± 0.2	19.2 ± 0.4	18.4 ± 0.4	18.4 ± 0.3	0.211	0.902
	Protein (%)	14.6 ± 0.1	14.9 ± 0.2	14.6 ± 0.2	14.9 ± 0.2	0.171	0.129
Females							
Unadjusted	Carbohydrate (%)	65.5 ± 0.3	66.6 ± 0.7	70.9 ± 0.5	70.6 ± 0.4	<0.001	<0.001
	Fat (%)	20.4 ± 0.2	19.1 ± 0.5	15.7 ± 0.4	16.0 ± 0.3	<0.001	<0.001
	Protein (%)	14.0 ± 0.1	14.4 ± 0.4	13.4 ± 0.2	13.3 ± 0.2	<0.001	<0.001
Model 1	Carbohydrate (%)	67.2 ± 0.2	66.8 ± 0.6	67.9 ± 0.5	67.9 ± 0.4	0.333	0.138
	Fat (%)	18.9 ± 0.2	18.8 ± 0.5	18.2 ± 0.4	18.3 ± 0.3	0.257	0.060
	Protein (%)	13.7 ± 0.1	14.3 ± 0.3	13.8 ± 0.2	13.7 ± 0.2	0.483	0.812
Model 2	Carbohydrate (%)	65.7 ± 0.6	65.2 ± 0.8	66.3 ± 0.7	66.2 ± 0.6	0.462	0.216
	Fat (%)	19.6 ± 0.5	19.5 ± 0.7	18.9 ± 0.6	19.1 ± 0.5	0.366	0.098
	Protein (%)	14.6 ± 0.3	15.2 ± 0.6	14.8 ± 0.4	14.7 ± 0.4	0.489	0.724
Model 3	Carbohydrate (%)	65.4 ± 0.3	65.3 ± 0.4	65.6 ± 0.4	65.5 ± 0.4	0.730	0.464
	Fat (%)	16.6 ± 0.5	16.3 ± 0.5	16.2 ± 0.4	16.3 ± 0.5	0.342	0.089
	Protein (%)	12.9 ± 0.3	13.2 ± 0.4	13.0 ± 0.3	12.9 ± 0.3	0.628	0.860

Values are means ± standard errors. Model 1: with adjustment for age; Model 2: Model 1 with adjustment for smoking status, alcohol consumption, physical activity level, and household income; and Model 3: Model 2 with adjustment for daily fiber, carbohydrate, protein, and fat intakes (except for models exploring individual macronutrients). Abbreviations: MHNW, metabolically healthy and of normal weight; MHO, metabolically healthy but obese; MANW, metabolically abnormal but of normal weight; MAO, metabolically abnormal and obese.

**Table 3 nutrients-11-00628-t003:** Associations between high or low macronutrient intakes and the obesity/metabolic risk phenotypes in males.

		Carbohydrate Intake		Fat Intake		Protein Intake	
		Low	High	Low	High	Low	High
Unadjusted	MHNW	1	1	1	1	1	1
	MHO	1.41 (1.04–1.92)	0.73 (0.58–0.92)	0.75 (0.59–0.97)	1.31 (0.94–1.83)	0.36 (0.04–3.41)	1.21 (0.81–1.80)
	MANW	0.58 (0.39–0.86)	1.39 (1.05–1.82)	1.39 (1.09–1.78)	0.58 (0.37–0.91)	0.37 (0.04–3.51)	0.71 (0.43–1.18)
	MAO	0.88 (0.66–1.17)	1.00 (0.80–1.24)	1.05 (0.84–1.32)	0.66 (0.47–0.93)	1.36 (0.27–6.86)	1.07 (0.75–1.53)
	*p* for trend	0.106	0.503	0.275	0.004	0.877	0.946
Model 1	MHNW	1	1	1	1	1	1
	MHO	1.26 (0.92–1.74)	0.83 (0.65–1.06)	0.89 (0.67–1.18)	1.17 (0.83–1.65)	0.39 (0.04–4.05)	1.13 (0.75–1.70)
	MANW	0.98 (0.65–1.48)	0.84 (0.65–1.11)	0.83 (0.64–1.09)	1.00 (0.61–1.62)	0.31 (0.04–2.54)	0.92 (0.55–1.53)
	MAO	1.08 (0.81–1.45)	0.82 (0.66–1.03)	0.88 (0.68–1.13)	0.81 (0.56–1.15)	1.28 (0.28–5.91)	1.18 (0.82–1.69)
	*p* for trend	0.573	0.056	0.193	0.313	0.949	0.466
Model 2	MHNW	1	1	1	1	1	1
	MHO	1.26 (0.92–1.73)	0.83 (0.65–1.07)	0.88 (0.67–1.17)	1.17 (0.83–1.65)	0.43 (0.04–4.34)	1.11 (0.73–1.69)
	MANW	0.89 (0.58–1.38)	0.86 (0.64–1.14)	0.83 (0.63–1.09)	0.95 (0.58–1.57)	0.32 (0.04–2.72)	0.81 (0.47–1.38)
	MAO	1.02 (0.76–1.36)	0.84 (0.67–1.05)	0.87 (0.67–1.20)	0.78 (0.54–1.13)	1.35 (0.30–6.15)	1.06 (0.74–1.52)
	*p* for trend	0.931	0.086	0.178	0.242	0.903	0.923
Model 3	MHNW	1	1	1	1	1	1
	MHO	1.08 (0.70–1.66)	1.24 (0.82–1.86)	0.96 (0.71–1.29)	1.18 (0.78–1.77)	0.58 (0.10–3.46)	1.14 (0.71–1.82)
	MANW	1.00 (0.55–1.81)	0.88 (0.55–1.42)	0.85 (0.60–1.21)	0.86 (0.50–1.49)	0.07 (0.001–9.47)	0.68 (0.36–1.27)
	MAO	0.93 (0.58–1.50)	0.85 (0.56–1.28)	0.91 (0.68–1.23)	0.66 (0.40–1.09)	2.02 (0.42–9.83)	0.96 (0.63–1.46)
	*p* for trend	0.804	0.416	0.415	0.122	0.765	0.641

Values are expressed as odds ratios (with 95% confidence intervals). Model 1: with adjustment for age; Model 2: Model 1 with adjustment for smoking status, alcohol consumption, physical activity level, and household income; and Model 3: Model 2 with adjustment for daily fiber, carbohydrate, protein, and fat intakes (except for models exploring individual macronutrients). High or low macronutrient intakes were those outside limits recommended for Korean adults (the recommended percentages of calories from carbohydrate, protein, and fat are 55–65, 7–20, and 15–30%, respectively). Abbreviations: MHNW, metabolically healthy and of normal weight; MHO, metabolically healthy but obese; MANW, metabolically abnormal but of normal weight; MAO, metabolically abnormal and obese.

**Table 4 nutrients-11-00628-t004:** Associations between high or low macronutrient intakes and the obesity/metabolic risk phenotypes in females.

		Carbohydrate Intake		Fat Intake		Protein Intake	
		Low	High	Low	High	Low	High
Unadjusted	MHNW	1	1	1	1	1	1
	MHO	0.72 (0.50–1.05)	1.03 (0.79–1.33)	1.26 (0.98-1.63)	0.76 (0.50–1.14)	1.15 (0.24–5.43)	1.28 (0.75–2.16)
	MANW	0.46 (0.32–0.65)	2.24 (1.81–2.77)	2.73 (2.26–3.30)	0.44 (0.30–0.63)	0.78 (0.23–2.70)	1.02 (0.68–1.51)
	MAO	0.45 (0.32–0.64)	2.10 (1.71–2.57)	2.18 (1.80–2.63)	0.38 (0.26–0.54)	2.30 (0.77–6.86)	0.94 (0.61–1.45)
	*p* for trend	<0.001	<0.001	<0.001	<0.001	0.259	0.846
Model 1	MHNW	1	1	1	1	1	1
	MHO	0.87 (0.60–1.26)	0.80 (0.60–1.07)	0.96 (0.72–1.29)	0.92 (0.60–1.42)	1.32 (0.27–6.37)	1.42 (0.83–2.42)
	MANW	0.87 (0.60–1.27)	1.12 (0.88–1.42)	1.25 (1.01–1.56)	0.85 (0.57–1.27)	1.26 (0.35–4.55)	1.44 (0.94–2.20)
	MAO	0.85 (0.59–1.22)	1.08 (0.87–1.35)	1.03 (0.83–1.28)	0.71 (0.49–1.04)	3.65 (0.97–13.72)	1.31 (0.83–2.07)
	*p* for trend	0.277	0.378	0.368	0.076	0.090	0.092
Model 2	MHNW	1	1	1	1	1	1
	MHO	0.85 (0.59–1.23)	0.80 (0.60–1.07)	0.96 (0.72–1.29)	0.91 (0.59–1.40)	1.35 (0.30–6.08)	1.37 (0.82–2.30)
	MANW	0.87 (0.59–1.26)	1.11 (0.87–1.42)	1.25 (1.00–1.56)	0.85 (0.57–1.27)	1.26 (0.34–4.60)	1.42 (0.93–2.16)
	MAO	0.84 (0.58–1.21)	1.07 (0.86–1.35)	1.02 (0.82–1.27)	0.71 (0.49–1.03)	3.69 (0.94–14.53)	1.29 (0.82–2.04)
	*p* for trend	0.256	0.444	0.415	0.068	0.097	0.110
Model 3	MHNW	1	1	1	1	1	1
	MHO	0.77 (0.41–1.46)	0.78 (0.41–1.46)	1.01 (0.70–1.47)	0.89 (0.51–1.53)	5.85 (1.13–30.31)	1.11 (0.60–2.07)
	MANW	0.69 (0.39–1.21)	1.01 (0.62–1.64)	1.15 (0.86–1.54)	0.80 (0.49–1.32)	1.63 (0.09–29.26)	1.47 (0.87–2.48)
	MAO	0.81 (0.43–1.54)	0.92 (0.59–1.43)	0.89 (0.63–1.25)	0.64 (0.38–1.09)	6.40 (0.97–42.06)	1.44 (0.81–2.54)
	*p* for trend	0.288	0.739	0.751	0.084	0.096	0.119

Values are expressed as odds ratios (with 95% confidence intervals). Model 1: with adjustment for age; Model 2: Model 1 with adjustment for smoking status, alcohol consumption, physical activity level, and household income; and Model 3: Model 2 with adjustment for daily fiber, carbohydrate, protein, and fat intakes (except for models exploring individual macronutrients). High or low macronutrient intakes were those outside limits recommended for the Korean adults (the recommended percentages of calories from carbohydrate, protein, and fat are 55–65, 7–20, and 15–30%, respectively). Abbreviations: MHNW, metabolically healthy and of normal weight; MHO, metabolically healthy but obese; MANW, metabolically abnormal but of normal weight; MAO, metabolically abnormal and obese.

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
