# Peer review of "Associations between Macronutrient Intakes and Obesity/Metabolic Risk Phenotypes: Findings of the Korean National Health and Nutrition Examination Survey"

_nutrients, 2019, doi:10.3390/nu11030628_

Round 1

Reviewer 1 Report

 The study by Ha-Na Kim and Sang-Wook Song reports on the association between macronutrient intakes and obesity/metabolic phenotypes with more than 7000 participants being analyzed and results being adjusted for age, smoking, alcohol consumption, physical activity, household income, and fiber intake. The authors found a positive association betwen metabolically healthy obese women and low protein intake. This is a well-written and interesting paper, and the inherent limitation of the study is adequately taken into account. I have several comments regarding this study.

Could the discrepancies between the results obtained in this study and other studies be explained, at least in part, by the origin of macronutrients in typically Korean food/ways of cooking and/or vitamins/minerals present in the Korean diet? Please discuss

Some dietary intervention studies with high-protein/low carbohydrate diet, when compared to normal protein/carbohydrate isocaloric diet found no effect between these diets on the blood parameters (see for instance Beaumont M. et al. Am J Clin Nutr 2017, 106:1005-1019; Westerterp-Plantenga MS et al. Br J Nutr 2012, 108:S105-S112), reinforcing the view that energy intake is central for fixing metabolic parameters.Please discuss on that point.

Do the authors have any information regarding the protein source (or major protein sources) in the participant's diet? For instance diet containing soy protein/associated isoflavones are well known to be able in several studies to lower systolic blood pressure when compared to diet containing animal protein. Please discuss.

Since protein are more satietogenic than carbohydrates and lipids (on the basis of equal energy intake), could low protein intake be associated with higher energy intake explaining the association found?

Author Response

Comments and Suggestions for Authors

 The study by Ha-Na Kim and Sang-Wook Song reports on the association between macronutrient intakes and obesity/metabolic phenotypes with more than 7000 participants being analyzed and results being adjusted for age, smoking, alcohol consumption, physical activity, household income, and fiber intake. The authors found a positive association betwen metabolically healthy obese women and low protein intake. This is a well-written and interesting paper, and the inherent limitation of the study is adequately taken into account. I have several comments regarding this study.

Could the discrepancies between the results obtained in this study and other studies be explained, at least in part, by the origin of macronutrients in typically Korean food/ways of cooking and/or vitamins/minerals present in the Korean diet? Please discuss

Answer) Thank you so much for your considerable comments. As you know and we mentioned in this manuscript, there has been only one study that investigated macronutrients intake and the combinations of obesity with metabolic abnormalities (as the obesity/metabolic risk phenotypes) [1], whereas there were several studies examined the associations between food intake (vegetables, fruits, grains, fish..) and the obesity/metabolic risk phenotypes, or macronutrients intake and obesity or health outcome [2-10]. Therefore, we already explained some mechanism about the discrepancies between the results and no association with macronutrient intakes and obesity/metabolic risk phenotypes (line 222-53). Furthermore, we hypothesized that carbohydrate, protein, or fat intakes are related to obesity/metabolic risk phenotypes in Korean adults. Therefore, we evaluated only the macronutrient intakes, not food type or diet pattern. As the reviewer recommended, we have added the contents of vitamins and minerals intake in the Limitation section (line 269-71).

Some dietary intervention studies with high-protein/low carbohydrate diet, when compared to normal protein/carbohydrate isocaloric diet found no effect between these diets on the blood parameters (see for instance Beaumont M. et al. Am J Clin Nutr 2017, 106:1005-1019; Westerterp-Plantenga MS et al. Br J Nutr 2012, 108:S105-S112), reinforcing the view that energy intake is central for fixing metabolic parameters.Please discuss on that point.

 Answer) As the reviewer recommended, we have added the contents in the fourth paragraph of the Discussion section (line 234-9).

Do the authors have any information regarding the protein source (or major protein sources) in the participant's diet? For instance diet containing soy protein/associated isoflavones are well known to be able in several studies to lower systolic blood pressure when compared to diet containing animal protein. Please discuss.

Answer) Thank you for the detailed suggestions. As we mentioned it in the Limitation section (line 267-9), the information was not used in this analysis due to insufficient data of the KNHANES.

Since protein are more satietogenic than carbohydrates and lipids (on the basis of equal energy intake), could low protein intake be associated with higher energy intake explaining the association found?

Answer) As the reviewer mentioned, we agree with your opinion, and had explained the mechanism about the significant finding of this study in the fifth paragraph of Discussion section (line 254-60).

References)

1.             Choi, J.; Se-Young, O.; Lee, D.; Tak, S.; Hong, M.; Park, S.M.; Cho, B.; Park, M. Characteristics of diet patterns in metabolically obese, normal weight adults (Korean National Health and Nutrition Examination Survey III, 2005). Nutrition, metabolism, and cardiovascular diseases : NMCD 2012, 22, 567-574, doi:10.1016/j.numecd.2010.09.001.

2.             Smith, H.A.; Gonzalez, J.T.; Thompson, D.; Betts, J.A. Dietary carbohydrates, components of energy balance, and associated health outcomes. Nutrition reviews 2017, 75, 783-797, doi:10.1093/nutrit/nux045.

3.             Kimokoti, R.W.; Judd, S.E.; Shikany, J.M.; Newby, P.K. Food intake does not differ between obese women who are metabolically healthy or abnormal. The Journal of nutrition 2014, 144, 2018-2026, doi:10.3945/jn.114.198341.

4.             Vitale, M.; Masulli, M.; Rivellese, A.A.; Babini, A.C.; Boemi, M.; Bonora, E.; Buzzetti, R.; Ciano, O.; Cignarelli, M.; Cigolini, M., et al. Influence of dietary fat and carbohydrates proportions on plasma lipids, glucose control and low-grade inflammation in patients with type 2 diabetes-The TOSCA.IT Study. European journal of nutrition 2016, 55, 1645-1651, doi:10.1007/s00394-015-0983-1.

5.             Perez-Martinez, P.; Alcala-Diaz, J.F.; Delgado-Lista, J.; Garcia-Rios, A.; Gomez-Delgado, F.; Marin-Hinojosa, C.; Rodriguez-Cantalejo, F.; Delgado-Casado, N.; Perez-Caballero, A.I.; Fuentes-Jimenez, F.J., et al. Metabolic phenotypes of obesity influence triglyceride and inflammation homoeostasis. European journal of clinical investigation 2014, 44, 1053-1064, doi:10.1111/eci.12339.

6.             Kimokoti, R.W.; Judd, S.E.; Shikany, J.M.; Newby, P.K. Metabolically Healthy Obesity Is Not Associated with Food Intake in White or Black Men. The Journal of nutrition 2015, 145, 2551-2561, doi:10.3945/jn.115.221283.

7.             Al-Khalidi, B.; Kimball, S.M.; Kuk, J.L.; Ardern, C.I. Metabolically healthy obesity, vitamin D, and all-cause and cardiometabolic mortality risk in NHANES III. Clinical nutrition (Edinburgh, Scotland) 2018, 10.1016/j.clnu.2018.02.025, doi:10.1016/j.clnu.2018.02.025.

8.             Kim, H.N.; Kim, S.H.; Eun, Y.M.; Song, S.W. Obesity with metabolic abnormality is associated with the presence of carotid atherosclerosis in Korean men: a cross-sectional study. Diabetology & metabolic syndrome 2015, 7, 68, doi:10.1186/s13098-015-0063-y.

9.             Parr, E.B.; Coffey, V.G.; Cato, L.E.; Phillips, S.M.; Burke, L.M.; Hawley, J.A. A randomized trial of high-dairy-protein, variable-carbohydrate diets and exercise on body composition in adults with obesity. Obesity (Silver Spring, Md.) 2016, 24, 1035-1045, doi:10.1002/oby.21451.

10.          Gutierrez-Repiso, C.; Soriguer, F.; Rojo-Martinez, G.; Garcia-Fuentes, E.; Valdes, S.; Goday, A.; Calle-Pascual, A.; Lopez-Alba, A.; Castell, C.; Menendez, E., et al. Variable patterns of obesity and cardiometabolic phenotypes and their association with lifestyle factors in the [email protected] study. Nutrition, metabolism, and cardiovascular diseases : NMCD 2014, 24, 947-955, doi:10.1016/j.numecd.2014.04.019.

Reviewer 2 Report

The paper is well written and the issue is interesting. However the design is strongly limiting. The limits of the study have strongly to be discussed.

what is the cause and what is the effect? This is the major problem of the cross-sectional design. Furthermore, one single 24h recall is not so informative.

Author Response

Comments and Suggestions for Authors

The paper is well written and the issue is interesting. However the design is strongly limiting. The limits of the study have strongly to be discussed.

what is the cause and what is the effect? This is the major problem of the cross-sectional design. Furthermore, one single 24h recall is not so informative.

Answer) Thank you for your constructive comments. As the reviewer mentioned, the cross-sectional design is deficient in determining causality. However, the design can discern how the variables are associated. Therefore, we had mentioned the content in the limitation part of the Discussion section (line 264-7). In terms of the dietary evaluation, a 24-h recall method has somewhat drawbacks, so we had tried to compensate for it and mentioned the contents in the limitation part (line 264-7).

Round 2

Reviewer 1 Report

 My comments have been adequately taken into account.